# The Calcineurin-FoxO-MuRF1 signaling pathway regulates myofibril integrity in cardiomyocytes

Hirohito Shimizu[1], Adam D Langenbacher[1], Jie Huang[1], Kevin Wang[1], Georg Otto[2], Robert Geisler[3], Yibin Wang[4,5], Jau-Nian Chen[1]*

[1]Department of Molecular, Cell and Developmental Biology, University of California, Los Angeles, Los Angeles, United States; [2]Genetics and Genomic Medicine, UCL Institute of Child Health, London, United Kingdom; [3]Institute of Toxicology and Genetics, Karlsruhe Institute of Technology, Karlsruhe, Germany; [4]Department of Anesthesiology, David Geffen School of Medicine, University of California, Los Angeles, Los Angeles, United States; [5]Department of Medicine and Physiology, David Geffen School of Medicine, University of California, Los Angeles, Los Angeles, United States

**Abstract** Altered $Ca^{2+}$ handling is often present in diseased hearts undergoing structural remodeling and functional deterioration. However, whether $Ca^{2+}$ directly regulates sarcomere structure has remained elusive. Using a zebrafish *ncx1* mutant, we explored the impacts of impaired $Ca^{2+}$ homeostasis on myofibril integrity. We found that the E3 ubiquitin ligase *murf1* is upregulated in *ncx1*-deficient hearts. Intriguingly, knocking down *murf1* activity or inhibiting proteasome activity preserved myofibril integrity, revealing a MuRF1-mediated proteasome degradation mechanism that is activated in response to abnormal $Ca^{2+}$ homeostasis. Furthermore, we detected an accumulation of the *murf1* regulator FoxO in the nuclei of *ncx1*-deficient cardiomyocytes. Overexpression of FoxO in wild type cardiomyocytes induced *murf1* expression and caused myofibril disarray, whereas inhibiting Calcineurin activity attenuated FoxO-mediated *murf1* expression and protected sarcomeres from degradation in *ncx1*-deficient hearts. Together, our findings reveal a novel mechanism by which $Ca^{2+}$ overload disrupts myofibril integrity by activating a Calcineurin-FoxO-MuRF1-proteosome signaling pathway.

DOI: https://doi.org/10.7554/eLife.27955.001

*For correspondence:
chenjn@mcdb.ucla.edu

## Introduction

The establishment and maintenance of rhythmic cardiac contractions require tightly regulated $Ca^{2+}$ signaling and intact contractile machinery. In the heart, a small amount of $Ca^{2+}$ enters cardiomyocytes upon stimulation by an action potential. This $Ca^{2+}$ influx induces the release of a larger amount of $Ca^{2+}$ from the sarcoplasmic reticulum (SR) resulting in an abrupt increase in cytosolic $Ca^{2+}$ levels and muscle contraction. The re-sequestration of $Ca^{2+}$ to the SR by SERCA2 and extrusion of $Ca^{2+}$ from the cell by NCX1 allows the muscle to relax (*Bers, 2002*). Abnormal $Ca^{2+}$ handling has been associated with cardiac diseases including heart failure and arrhythmia in humans and animal models (*Luo and Anderson, 2013*) and structurally defective myofibrils are also often observed in diseased hearts (*Lopes and Elliott, 2014*). However, whether or not there is a causal relationship between abnormal $Ca^{2+}$ handling and myofibril disarray in diseased myocytes has not yet been established.

The RING finger protein MuRF1 (also known as TRIM63) is a muscle-specific E3 ubiquitin protein ligase involved in the regulation of muscle turnover in normal physiology and under pathological conditions. MuRF1 acts on several sarcomeric target proteins, tagging them with polyubiquitin

chains for proteasome-dependent degradation (*Kedar et al., 2004*; *Clarke et al., 2007*; *Cohen et al., 2009*; *Mearini et al., 2010*). Through this mechanism, MuRF1 regulates normal sarcomere protein turnover and removes misfolded and/or damaged proteins in skeletal and cardiac muscles (*Lyon et al., 2013*; *Pagan et al., 2013*; *Willis et al., 2014*). *Murf1* expression is elevated under muscle catabolic conditions and overexpression of *murf1* in the heart results in a thin ventricular wall and a rapid transition to heart failure upon transaortic constriction, suggesting that MuRF1 is a major player in muscle catabolic processes (*Bodine et al., 2001*; *Labeit et al., 2010*; *Baehr et al., 2011*; *Files et al., 2012*; *Gomes et al., 2012*; *Bodine and Baehr, 2014*). Conversely, knockout of *murf1* promotes resistance to muscle atrophy and an exaggerated hypertrophic response to pressure overload (*Willis et al., 2007*; *Willis et al., 2009a*; *Willis et al., 2009b*). In humans, patients with specific *murf1* gene variants develop hypertrophic cardiomyopathy at a younger age (*Chen et al., 2012*; *Su et al., 2014*), revealing a pathological role for MuRF1 in the progression of cardiac diseases.

In skeletal muscles, the Forkhead box O (FoxO) transcription factor family serves as a nodal point controlling muscle degradation via the regulation of MuRF1 expression. Under catabolic conditions, the PI3K-Akt pathway is suppressed and hypophosphorylated FoxO translocates into the nucleus causing *murf1* induction and muscle atrophy (*Lecker et al., 2004*; *Waddell et al., 2008*). Conversely, upon IGF stimulation, the phosphorylation of FoxO by activated AKT sequesters FoxO in the cytoplasm, resulting in reduced *murf1* expression and an increase in myocyte mass (*Sacheck et al., 2004*; *Stitt et al., 2004*). Similarly, an AKT-FoxO-mediated suppression of *murf1* expression in response to insulin has been noted in cardiac muscles (*Skurk et al., 2005*; *Paula-Gomes et al., 2013*).

In this study, we used the zebrafish *tremblor/ncx1h* mutant to explore the regulatory relationship between $Ca^{2+}$ homeostasis and the maintenance of cardiac muscle integrity. We have previously shown that *ncx1h* (also known as *slc8a1a*) encodes a cardiac specific sodium-calcium exchanger 1 (NCX1) in zebrafish and that the *tremblor* (*tre*) mutant lacks functional NCX1h (*Langenbacher et al., 2005*). NCX1 is a primary $Ca^{2+}$ efflux mechanism in cardiomyocytes (*Ottolia et al., 2013*), and consistent with this important role in $Ca^{2+}$ homeostasis, cytosolic $Ca^{2+}$ levels are increased and cyclic $Ca^{2+}$ transients are abolished in *tre/ncx1h* cardiomyocytes resulting in fibrillation-like chaotic cardiac contractions (*Ebert et al., 2005*; *Langenbacher et al., 2005*; *Shimizu et al., 2015*). Like NCX1-/- mice, *tre/ncx1h* zebrafish hearts also develop severe myofibril disarray (*Koushik et al., 2001*; *Wakimoto et al., 2003*; *Ebert et al., 2005*), suggesting that a conserved molecular link exists between aberrant $Ca^{2+}$ handling and myofibril disarray. From a microarray analysis, we found that the expression of *murf1* is significantly upregulated in *ncx1h*-deficient hearts. This MuRF1 upregulation was responsible for the myofibril disarray in *ncx1h*-deficient hearts, and normal cardiac myofibrils could be restored by genetic and pharmacological manipulation of MuRF1 or proteasome activity. We also found that elevated intracellular $Ca^{2+}$ levels enhanced *murf1* expression via activation of Calcineurin signaling, which dephosphorylates the *murf1* transcriptional regulator FoxO, leading to its nuclear translocation. Our findings reveal a novel signaling pathway in which $Ca^{2+}$ homeostasis modulates the integrity of cardiac muscle structure via *murf1* regulation.

## Results and discussion

### NCX1 is required for the maintenance of myofibril integrity in cardiomyocytes

Zebrafish *ncx1h* mutant embryos lack functional NCX1 in myocardial cells resulting in aberrant $Ca^{2+}$ homeostasis and a fibrillating heart (*Ebert et al., 2005*; *Langenbacher et al., 2005*; *Shimizu et al., 2015*). Similar to the myofibril phenotype observed in NCX1-/- mice, sarcomeres in zebrafish *ncx1h* mutant cardiomyocytes are damaged (*Koushik et al., 1999*; *Wakimoto et al., 2003*; *Ebert et al., 2005*). To investigate whether NCX1 activity affects the assembly or the maintenance of sarcomeres in myocardial cells, we examined the distribution of α-actinin protein. In striated muscles, α-actinin is localized to the Z-line and is a good marker for assessing sarcomere structure. We found that α-actinin is organized into a periodic banding pattern in both wild type and *ncx1h* mutant cardiomyocytes at 30 hpf (*Figure 1A*), suggesting that sarcomere assembly is initiated properly in the absence of NCX1 activity. Interestingly, the sarcomeres degenerate in *ncx1h* mutant cardiomyocytes a day later

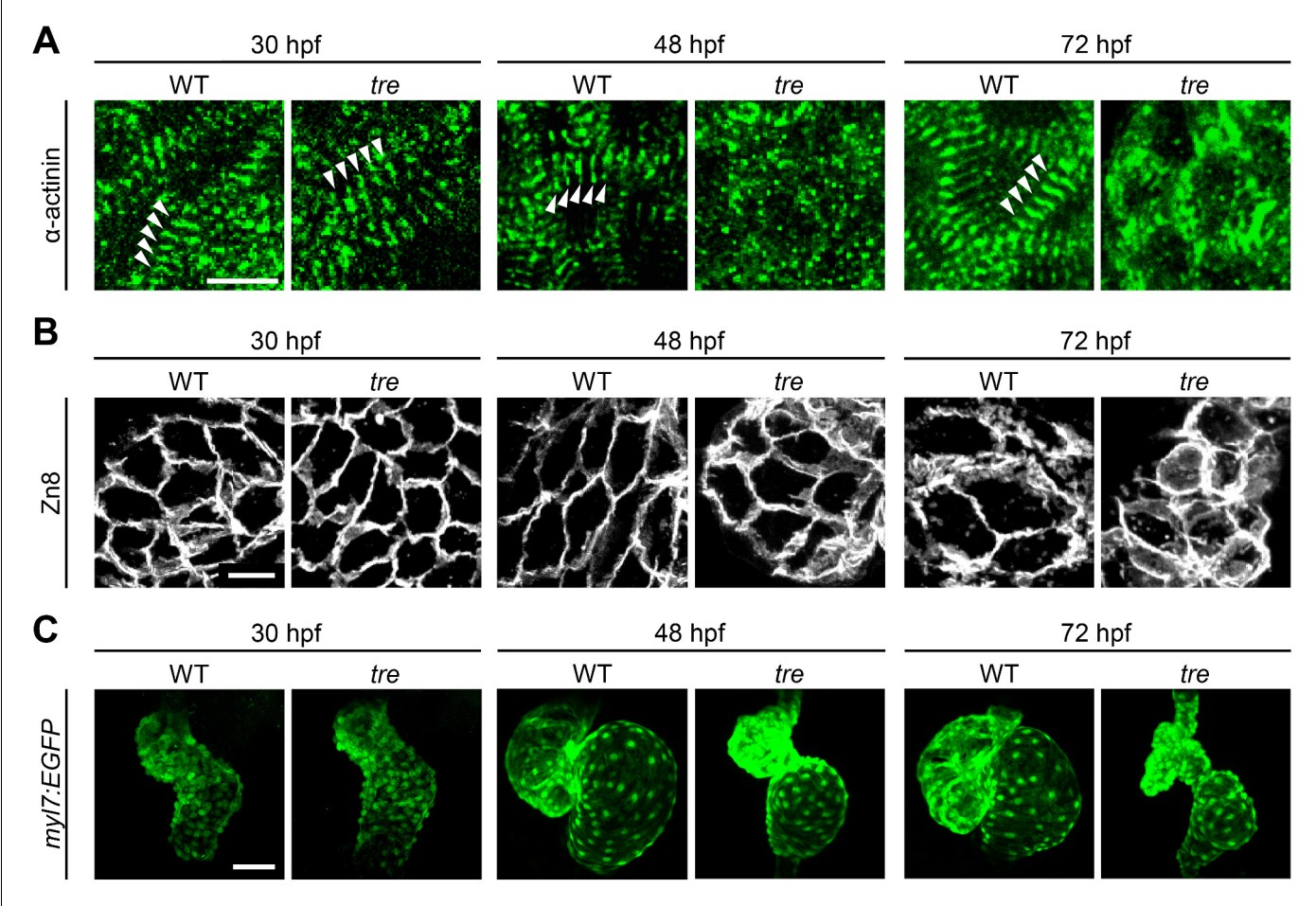

**Figure 1.** Disorganized myofibril structure in *tre/ncx1h* cardiomyocytes. Wild type (WT) and *tre/ncx1h* (*tre*) mutant hearts at 30, 48 and 72 hpf. (**A**) Zebrafish hearts stained for α-actinin to visualize Z-lines. At 30 hpf, periodic α-actitin staining was observed in wild type and *tre* hearts (arrowheads). By 48 hpf, sarcomeres are disassembled in *tre* hearts. Scale bar, 10 μm. (**B**) The cell shape of cardiomyocytes was visualized by Zn8 staining. Scale bar, 10 μm. (**C**) Embryonic fish hearts were visualized by GFP expression in the *myl7:EGFP* transgenic background. Note that *tre* hearts become dysmorphic after two days of development. Scale bar, 50 μm.

DOI: https://doi.org/10.7554/eLife.27955.002

resulting in a sporadic distribution of α-actinin (*Figure 1A*). Zebrafish myocardial cells of the outer curvature normally assume an elongated, flat shape by two days of development (*Auman et al., 2007*; *Cavanaugh et al., 2015*). However, *ncx1h* mutant cardiomyocytes fail to elongate (*Figure 1B*) and both atrial and ventricular chambers become dysmorphic (*Figure 1C*), indicating a requirement for NCX1 activity in the maintenance of myofibril integrity and cardiac chamber morphology.

## Elevated MuRF1 expression in *ncx1h*-deficient hearts

To explore molecular pathways by which NCX1 influences myofibril integrity, we isolated hearts from wild type and *ncx1h* mutant embryos and compared their gene expression profiles. We found that the expression of Muscle Ring-finger protein-1 (MuRF1, also known as TRIM63) is significantly elevated in *ncx1h* mutant hearts. There are two highly homologous *murf1* genes in zebrafish (*murf1a/trim63a* and *murf1b/trim63b*) (*Macqueen et al., 2014*). Phylogenetic analysis showed that zebrafish *murf1a* and *murf1b* cluster with other vertebrate *murf1* genes (*Figure 2A*). Both genes span a single exon encoding peptides highly homologous to each other and to their mammalian orthologs (*Figure 2B*) (*Postlethwait, 2007*) and are expressed in striated muscles (*Figure 2C*) (*Willis and Patterson, 2006*). In situ hybridization and quantitative RT-PCR analyses further confirmed that both *murf1a* and *1b* are upregulated in *ncx1* mutant hearts (*Figure 3*).

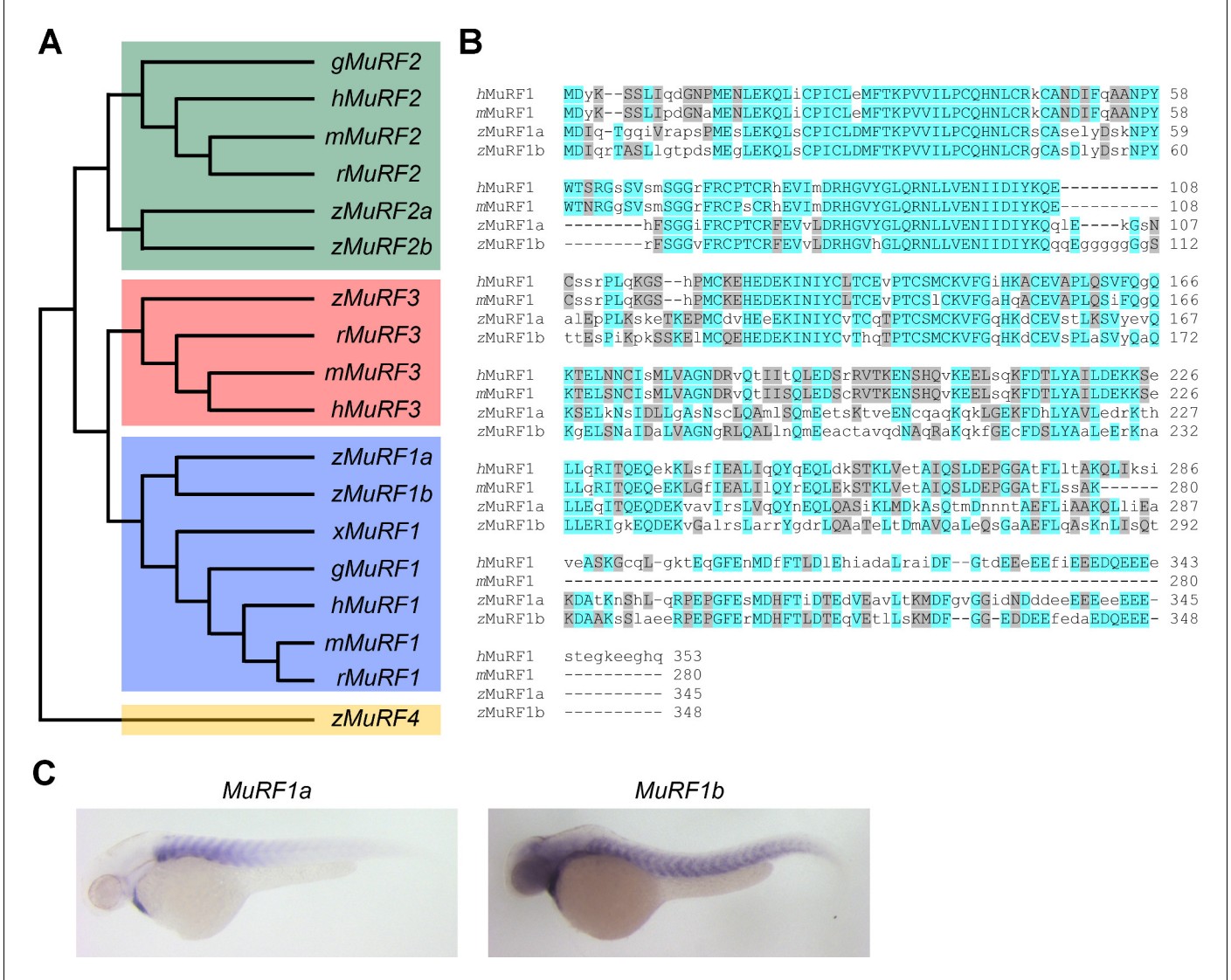

**Figure 2.** Zebrafish *murf1* genes. (A) Phylogenetic tree of vertebrate *murf1*, *2, 3* and *4* (also known as *trim63*, *55*, *54* and *101*, respectively). The tree was constructed using ClustalX with the neighbor-joining method. Zebrafish (z), Human (h), mouse (m), rat (r), chick (g), frog (x). (B) Alignment of *murf1* genes from human, mouse and zebrafish. Blue boxes highlight identical amino acids and grey boxes indicate similar residues. (C) Whole-mount in situ hybridization demonstrating the expression patterns of *murf1a* and *murf1b* in the zebrafish embryo.

DOI: https://doi.org/10.7554/eLife.27955.003

## MuRF1 regulates myofibril integrity in cardiomyocytes

Elevated *murf1* expression is associated with muscle atrophy and can induce the breakdown of myofibrils in cultured cardiomyocytes (*Kedar et al., 2004*). We thus asked whether MuRF1 overexpression in the heart is sufficient to induce cardiomyopathy. To this end, we generated a transgenic fish, *myl7:MuRF1a-IRES-GFP*, in which MuRF1 and a GFP reporter are expressed under the control of the cardiac-specific *myl7* promoter (*Figure 4A*). As shown in *Figure 4B*, *murf1a* expression is upregulated in *myl7:MuRF1a-IRES-GFP* transgenic hearts. Interestingly, α-actinin failed to maintain a striated pattern in cardiomyocytes of *myl7:MuRF1a-IRES-GFP* embryos (*Figure 4C*), demonstrating that overexpression of MuRF1 leads to sarcomere disassembly in the heart. Consequently, MuRF1-overexpressing hearts become dilated (*Figure 4D*) and their cardiac function is compromised (*Videos 1* and *2*). The atrial fractional shortening of *myl7:MuRF1a-IRES-GFP* hearts was reduced by approximately 10% and ventricular fractional shortening by ~15% compared to non-transgenic siblings and

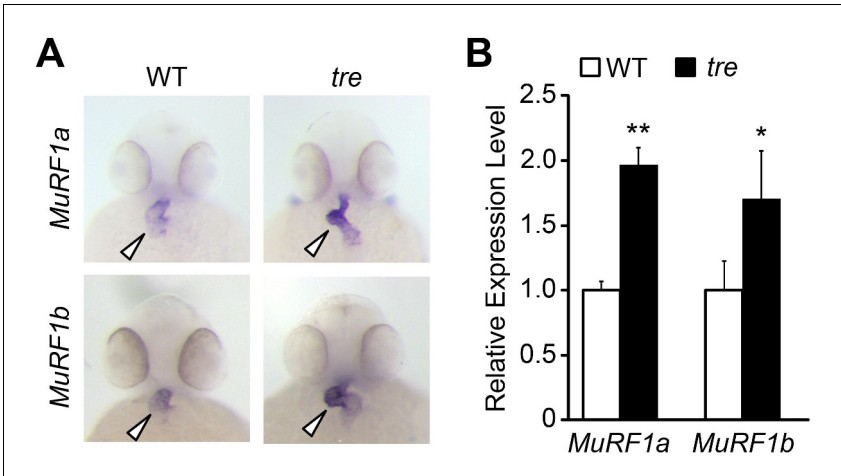

**Figure 3.** Upregulation of MuRF1 in *tre/ncx1h* deficient hearts. (**A**) In situ hybridization analysis shows a significant increase of *murf1a* and *murf1b* expression in *tre* hearts. Arrowheads point to the heart. (**B**) Quantitative RT-PCR analysis shows an upregulation of *murf1a* and *murf1b* in *tre* hearts. * $p<0.05$; ** $p<0.01$.
DOI: https://doi.org/10.7554/eLife.27955.004

the heart rate was also reduced by 10% (*Figure 4E*). Together, these findings demonstrate that overexpression of MuRF1 is sufficient to disrupt myofibril structure and impair cardiac function in vivo.

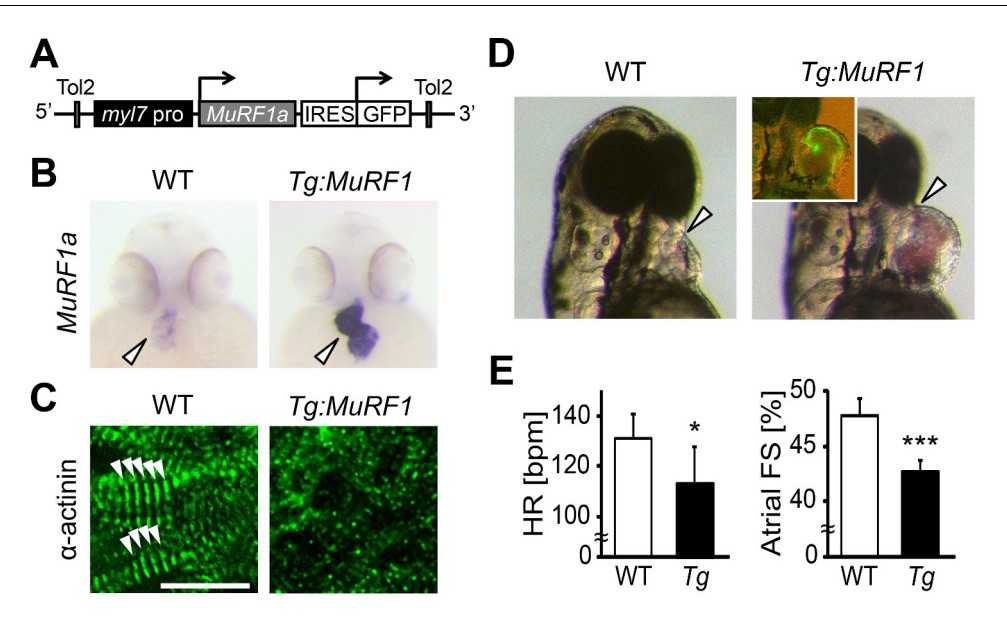

**Figure 4.** Upregulation of MuRF1a leads to myofibril disarray. (**A**) Schematic representation of the construct that drives cardiac-specific MuRF1a expression. (**B**) *Murf1a* expression is upregulated in the 2-day-old *myl7:MuRF1a-IRES-GFP* heart (right panel) compared to the wild type heart (left panel). (**C**) α-actinin staining in 3-day-old wild type (left panel) and transgenic (right panel) cardiomyocytes. Note that sarcomeres are disassembled in *myl7: MuRF1a-IRES-GFP* cardiomyocytes. (**D**) Live images of wild type and *Tg(myl7:MuRF1a-IRES-GFP)* fish at 72 hpf (left panels). Transgenic hearts are GFP positive and become dilated (inset). (**E**) Heart rate (HR) and atrial fractional shortening (FS) in wild type and *myl7:MuRF1a-IRES-GFP* embryos at 72 hpf. *p<0.05; ***p<0.001.
DOI: https://doi.org/10.7554/eLife.27955.005

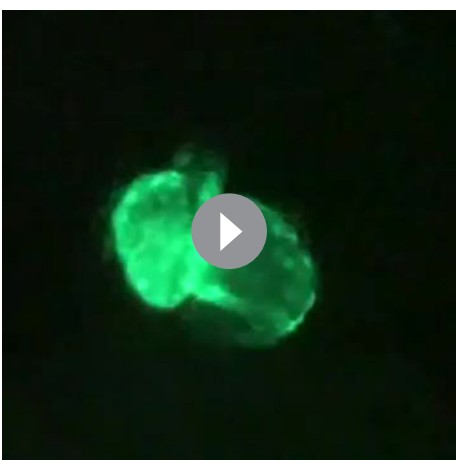

**Video 1.** Three-day-old *myl7:EGFP* transgenic heart.
DOI: https://doi.org/10.7554/eLife.27955.006

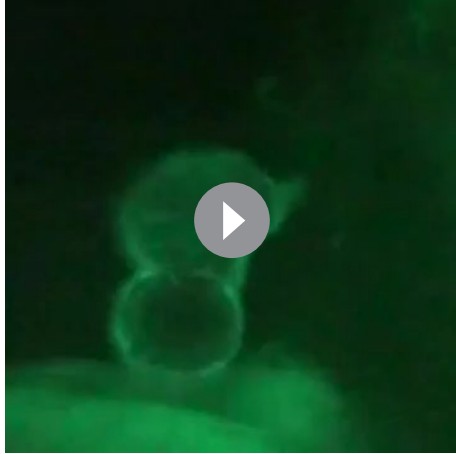

**Video 2.** Three-day-old *myl7:MuRF1a-IRES-GFP* transgenic heart.
DOI: https://doi.org/10.7554/eLife.27955.007

## Blocking MuRF1-induced protein degradation preserves myofibril integrity in *ncx1h* mutant hearts

MuRF1's upregulation upon loss of *ncx1h* activity, along with its established function as a muscle-specific E3 ubiquitin protein ligase that targets sarcomeric proteins for proteasome degradation, make it a good candidate for the cause of the myofibril disarray present in *ncx1h* deficient hearts. If MuRF1 upregulation indeed causes sarcomere disassembly, one would predict that blocking MuRF1 activity or its downstream protein degradation pathway might ameliorate the myofibril defects in *ncx1* mutant hearts. Since both *murf1a* and *murf1b* are upregulated in *ncx1h* deficient hearts, we knocked down these genes simultaneously. Western blot analysis showed that *murf1a/murf1b* morpholino knockdown reduced overall MuRF1 protein levels by 27% (*Figure 5C*). Interestingly, even this modest level of reduction in MuRF1 expression improved sarcomere integrity in *ncx1h* deficient hearts; ~80% of *ncx1h/murf1a/murf1b* triple-deficient embryos had intact sarcomeres (n = 24), a significant increase compared to *ncx1h* mutant hearts (~35%, n = 21; p<0.001) (*Figure 5A,B*). Similarly, treatment with the proteasome inhibitor MG132 suppressed the myofibril disarray caused by NCX1 deficiency. Approximately 72% of MG132-treated *ncx1h* mutants had a striated α-actinin pattern, indicating the presence of intact sarcomeres (n = 18; p<0.001) (*Figure 5A,B*), suggesting that upregulation of MuRF1 induces myofibril degradation via a proteasome-dependent mechanism in *ncx1h*-deficient cardiomyocytes. Prior studies have implied a connection between cardiac contraction and myofibril integrity (*Berdougo et al., 2003*; *Auman et al., 2007*; *Nishii et al., 2008*; *Yang et al., 2014*). Interestingly, *ncx1h/murf1a/murf1b* triple-deficient hearts never establish coordinated contractions (*Video 3*), demonstrating that sarcomere integrity can be uncoupled from the loss of cardiac contractions in the context of aberrant $Ca^{2+}$ handling-induced heart failure.

## $Ca^{2+}$ induces MuRF1a expression

Our study showed that while *ncx1h* mutant hearts suffer from $Ca^{2+}$ handling defects and never establish normal $Ca^{2+}$ cycles or heartbeats, the initial assembly of sarcomeres proceeds properly. Since the $Ca^{2+}$ handling defects precede the breakdown of sarcomeres, we hypothesized that $Ca^{2+}$ overload induces *murf1* expression in cardiomyocytes and thereby leads to sarcomere disassembly and cardiomyopathy. To examine this hypothesis, we isolated a 6.9 kb genomic fragment upstream of *murf1a*, MuRF1a (−6906). Transgenic analysis showed that this genomic fragment was sufficient to drive GFP expression in cardiac and skeletal muscles (*Figure 6A and B*), a pattern resembling the endogenous *murf1a* expression pattern (*Figure 2C*), indicating that critical regulatory elements are present in this genomic fragment. We then created a MuRF1a (−6906) Luciferase reporter construct to test whether this *murf1* upstream regulatory element is responsive to $Ca^{2+}$ signaling. We transfected the MuRF1a (−6906) Luciferase reporter into HEK293T cells and induced $Ca^{2+}$ flux by treatment with the $Ca^{2+}$ ionophore A23187. Interestingly, the luciferase activity was significantly

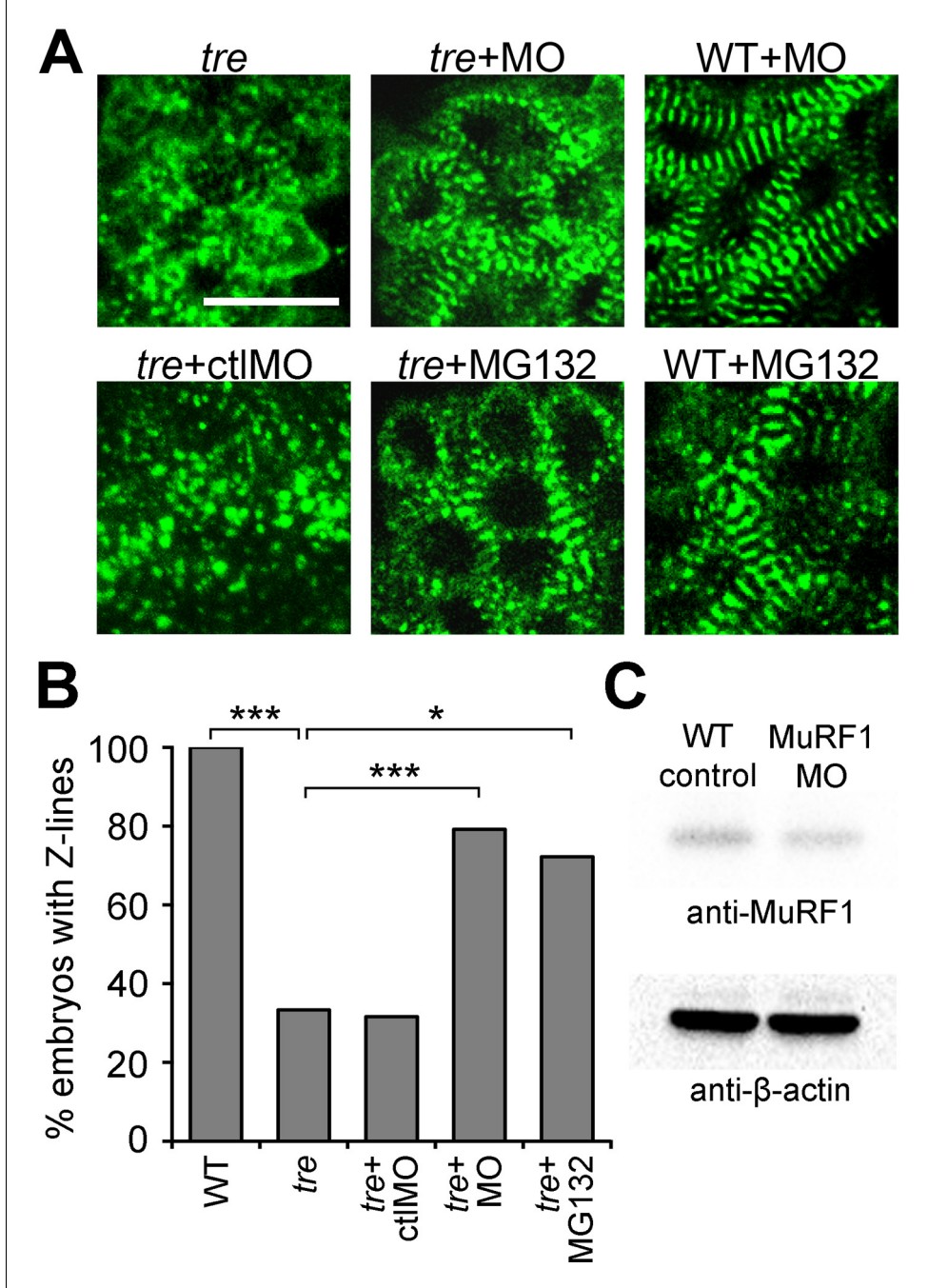

**Figure 5.** Blocking MuRF1-mediated proteasome degradation preserves myofibril integrity in *tre/ncx1* deficient hearts. (**A**) Z-lines were visualized by α-actinin staining. By 72 hpf, sarcomeres are disassembled in hearts of uninjected (*tre*) and control morpholino-injected (*tre* +ctlMO) *tremblor* embryos. *Murf1a/murf1* b knockdown does not affect sarcomere integrity in wild type embryos (WT +MO), but prevents sarcomere degradation in *tre* (*tre* +MO). Similarly, treatment with a proteasome inhibitor, MG132, preserves myofibril integrity in *tre* cardiomyocytes (*tre* +MG132). Scale bar, 10 µm. (**B**) Graph shows % of embryos with periodic α-actinin staining at 72 hpf. (**C**) Western blot detecting MuRF1 and β-actin proteins in uninjected control (WT control) and *murf1a/ murf1* b knockdown (MuRF1 MO) embryos. Chi-squared test, *p<0.05; **p<0.01; ***p<0.001.
DOI: https://doi.org/10.7554/eLife.27955.008

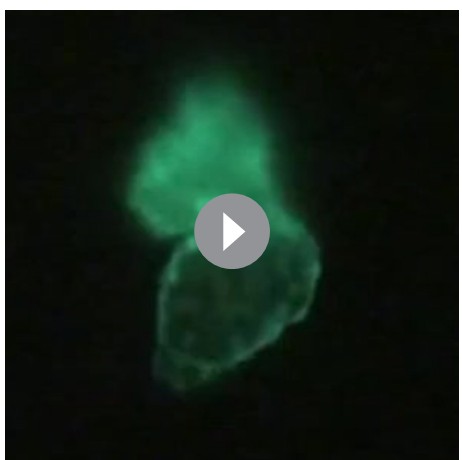

**Video 3.** Two-day-old *ncx1h/murf1a/murf1b* triple deficient heart.
DOI: https://doi.org/10.7554/eLife.27955.009

enhanced by A23187 induction (*Figure 6C*), demonstrating that *murf1* transcription is sensitive to intracellular $Ca^{2+}$ levels.

We next tested the expression patterns of a series of deletion constructs of MuRF1a (−6906) and found that the 638 bp region immediately upstream of the transcription initiation site, MuRF1a (−638), was sufficient to drive reporter gene expression in cardiac and skeletal muscles (*Figure 6A*). MuRF1a (−638)-Luc also displayed enhanced luciferase activity upon A23187 induction at levels comparable to MuRF1a (−6906)-Luc (*Figure 6D*), indicating that the 638 bp *murf1a* proximal region is sufficient to direct $Ca^{2+}$-mediated *murf1* transcription.

## $Ca^{2+}$ regulates MuRF1 expression via the Calcineurin-FoxO signaling pathway

Calmodulin-dependent protein kinase II (CaMKII) and the calmodulin-dependent protein phosphatase calcineurin (Cn) are two major transducers of $Ca^{2+}$ signals in cardiomyocytes (*Heineke and Molkentin, 2006*; *Maillet et al., 2013*). We asked whether either of these pathways is involved in the regulation of *murf1* gene expression. We treated MuRF1a (−638)-Luc-transfected HEK293T cells with either KN62, a chemical inhibitor of CaMKII, or FK506, an inhibitor of Cn. KN62 treatment did not have a significant impact on MuRF1a (−638)-driven expression, but FK506 treatment potently attenuated the A23187-induced increase of MuRF1a (−638)-Luc reporter activity (*Figure 7A*), suggesting that MuRF1 expression is regulated by a Cn-mediated mechanism. This interpretation is further supported by the observations that Cn overexpression enhances A23187-induced *murf1* reporter activity and that the A23187-induced MuRF1 expression is blunted by overexpression of a dominant negative form of Cn (DN-Cn) (*Figure 7B*) (*Kahl and Means, 2004*).

We next explored the potential molecular mechanisms by which Cn influences MuRF1 expression. We found multiple putative FoxO binding sites located the minimal regulatory regions of zebrafish *murf1a/b* and within the 1 kb region immediately upstream of the transcription initiation sites of the frog, mouse and human *murf1* genes (*Figure 7C*). Since FoxO is a downstream mediator of Cn signaling (*Hudson and Price, 2013*) and is involved in the regulation of MuRF1 in skeletal muscles (*Stitt et al., 2004*; *Waddell et al., 2008*), we examined the possibility that FoxO mediates Cn's regulation of MuRF1 expression in cardiomyocytes. There are seven *foxo* genes in zebrafish (*Wang et al., 2009*), all of which are expressed in the developing heart (*Figure 7D* and *Figure 8*). When co-transfected with the MuRF1a(−638)-Luc reporter, all zebrafish *foxo* genes tested were capable of enhancing the A23187-induced increase in MuRF1a(−638) luciferase activity (*Figure 9*). FoxO3a promoted strong *murf1a* promoter activity (*Figure 9*) and was used for the remainder of our analyses. We found that FoxO3a enhanced MuRF1a(−638)-Luc reporter activity in a dose-dependent manner (*Figure 7E*) whereas overexpression of a dominant-negative form of FoxO (DN-foxO), which lacks the transactivation domain but harbors an intact DNA binding domain (*Medema et al., 2000*; *van den Heuvel et al., 2005*), abrogated the A23187-induced MuRF1a(−638)-Luc activity (*Figure 7F*). We next asked whether FoxO mediates Cn signaling to control MuRF1 expression. We found that cotransfection of Cn and FoxO3a enhances *murf1a* promoter activity (*Figure 7B,E*) and that FoxO could no longer induce MuRF1a expression in the presence of a dominant negative form of Cn (*Figure 7B,E*), demonstrating that $Ca^{2+}$ influences MuRF1 expression via the Cn-FoxO signaling axis.

## Cn and FoxO regulate MuRF1 expression in the heart

Based on our finding that a Cn-FoxO-MuRF1 regulatory pathway is activated in response to elevated intracellular $Ca^{2+}$ levels in cultured cells, we explored the significance of the Cn-FoxO-MuRF1

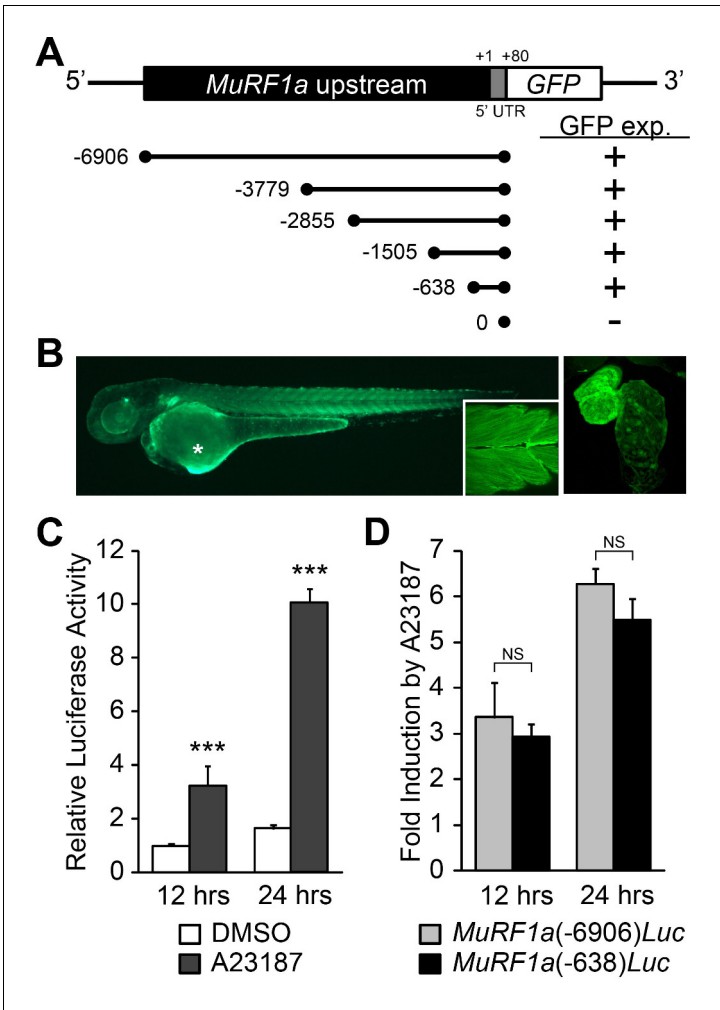

**Figure 6.** Identification of a MuRF1a regulatory element. (**A**) Schematic representation of *murf1a* reporter constructs. + denotes the presence of GFP expression in the heart and somites. (**B**) A MuRF1a (−6906)-GFP transgenic embryo exhibits GFP expression in the heart and the somites (inset shows higher magnification image of the somites). The right panel shows a higher magnification image of the heart. The asterisk denotes auto-fluorescence from the yolk. (**C**) A23187 treatment induces luciferase activity driven by the MuRF1a (−6906) promoter. Values on the y-axis represent the luciferase activity relative to cells treated with DMSO for 12 hr. (**D**) Comparison of $Ca^{2+}$ responsiveness between the MuRF1a (−638) and MuRF1a (−6906) promoters. Values on the y-axis represent the fold increase in luciferase activity in response to A23187 treatment compared to DMSO-treated cells at each time point. ***$p<0.001$; NS, not significant.

DOI: https://doi.org/10.7554/eLife.27955.010

pathway in the regulation of myofibril integrity in myocardial cells in vivo. The subcellular localization of FoxO is controlled by its phosphorylation status (*Huang and Tindall, 2007*). We reasoned that the $Ca^{2+}$ extrusion defect in *ncx1h* mutant hearts could activate Cn resulting in the dephosphoryla-tion and nuclear translocation of FoxO. To assess the subcellular localization of FoxO in cardiomyo-cytes, we injected the *myl7:FLAG-foxo3a-IRES-EGFP* plasmid into zebrafish embryos and used the FLAG-epitope as a proxy to assess the localization of FoxO. Indeed, while FoxO was primarily sequestered in the cytoplasm of cardiomyocytes in wild type zebrafish hearts, FoxO protein was enriched in the nuclei of *ncx1h* mutant cardiomyocytes (*Figure 10A*). This nuclear accumulation of FoxO correlated with the increased MuRF1 expression in *ncx1h* mutant hearts (*Figure 3*). In addition, we found that MuRF1 expression could also be induced in the heart by overexpression of FoxO3a or a constitutively active form of FoxO3a in which three phosphorylation sites were replaced by with alanines (CA-FoxO3a: T29A, S236A, S299A) (*Figure 10B*) (*Brunet et al., 1999*). Conversely,

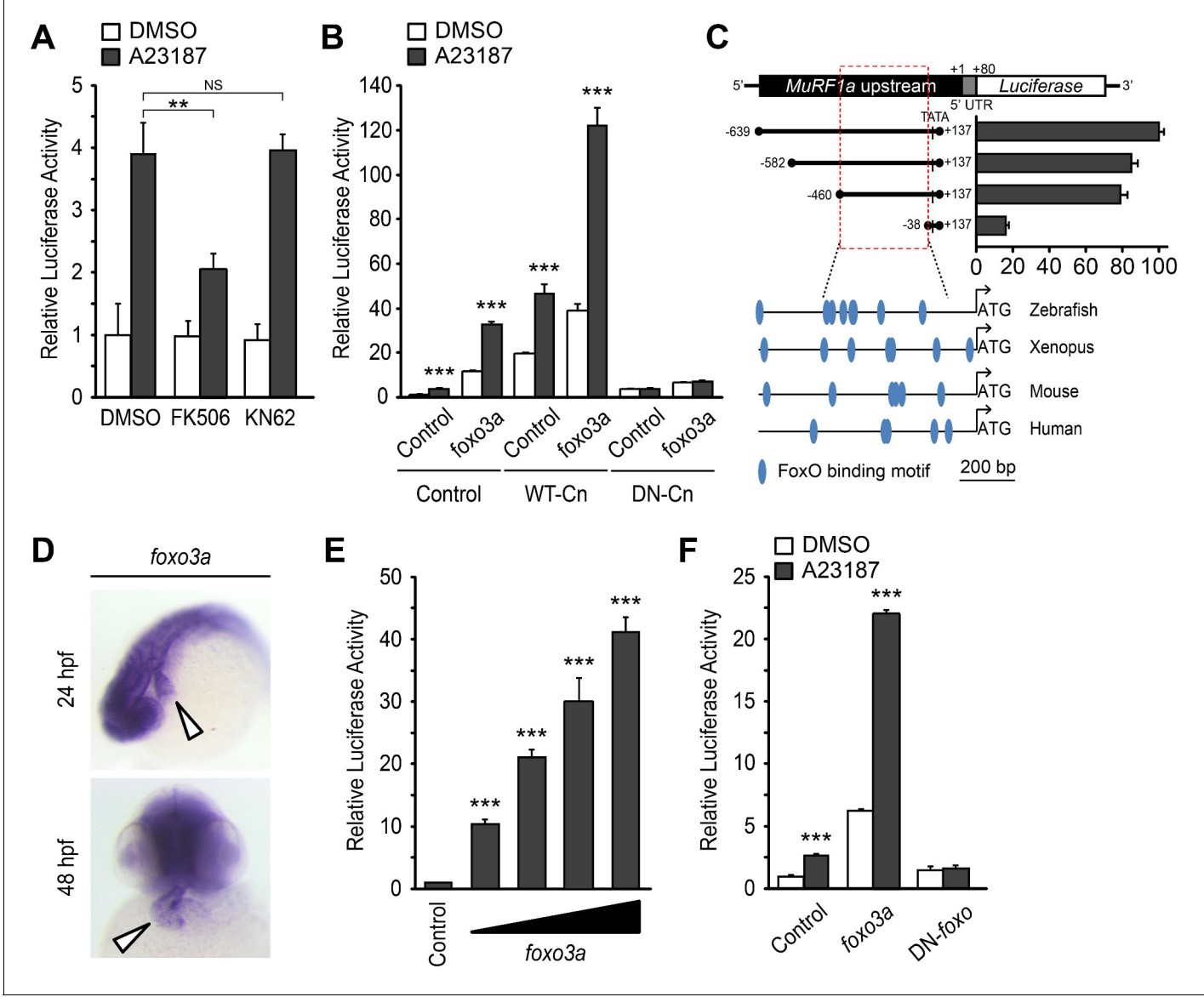

**Figure 7.** A Ca$^{2+}$-Cn-FoxO signaling pathway regulates MuRF1 expression. (**A**) HEK293T cells were transiently transfected with the MuRF1a (−638) luciferase reporter construct. Cells were incubated with FK506 or KN62 before DMSO or A23187 treatment. (**B**) Luciferase assay of the MuRF1a (−638) reporter cotransfected with *foxo3a*, wild type calcineurin (WT-Cn) or dominant-negative calcineurin (DN-Cn). (**C**) Diagram of the MuRF1a (−638) reporter serial deletion constructs generated for this study. The bar graph (right) shows the luciferase activity of each reporter construct relative to that of the empty expression plasmid. The red dotted box indicates the minimal *cis*-regulatory element of MuRF1a. The lower diagrams represent an alignment of the zebrafish, *Xenopus*, mouse and human MuRF1 promoters. Blue circles indicate putative FoxO binding sites (**D**) Whole-mount in situ hybridization detects *foxo3a* expression in the zebrafish heart. White arrowheads point to the heart. (**E**) HEK293T cells were transfected with the MuRF1a (−638) luciferase reporter and *foxo3a* expression plasmid. (**F**) HEK293T cells were transfected with the MuRF1a (−638) reporter plasmid and either a wild type or dominant negative foxo3a expression plasmid. Values on the y-axis are expressed relative to the luciferase activity of DMSO treated cells. \*\*p<0.01; \*\*\*p<0.001; NS, not significant.

DOI: https://doi.org/10.7554/eLife.27955.011

pharmacological inhibition of Cn activity by treatment with FK506 or overexpression of DN-FoxO blunted MuRF1 expression in *ncx1h* mutant embryos (*Figure 10B*). Finally, we used α-actinin as a proxy to examine whether the correlation between FoxO and MuRF1 expression translates to the preservation of sarcomere structure. We found that overexpression of CA-FoxO3a in wild type embryos resulted in a sporadic α-actinin distribution in cardiomyocytes that resembled the

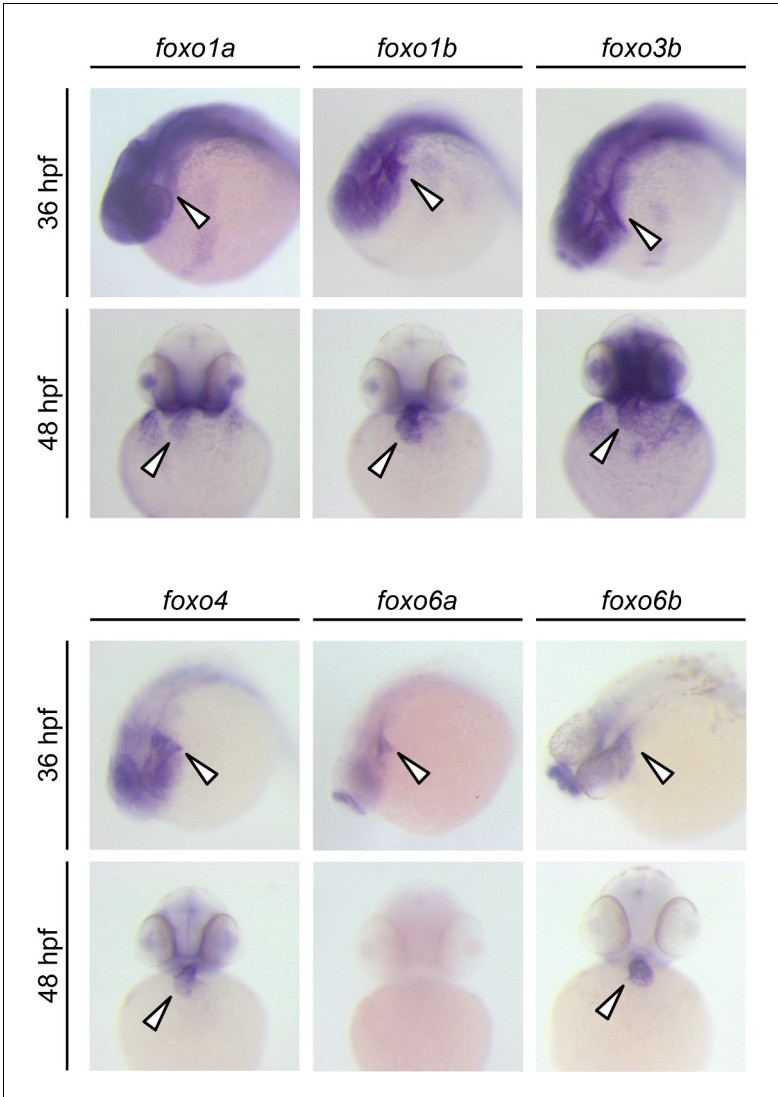

**Figure 8.** Expression patterns of zebrafish *foxo* genes. Whole-mount in situ hybridization analysis showing *foxo* expression in zebrafish embryos. While *foxo6a* expression is diminished by 48 hpf, all the other *foxo* genes examined (*foxo1a*, *1b*, *3b*, *4*, *6a* and *6b*) are persistently expressed in the heart. Arrowheads point to the heart.
DOI: https://doi.org/10.7554/eLife.27955.012

phenotype observed in *ncx1h* mutant hearts whereas overexpression of DN-FoxO restored a striated α-actinin pattern in *ncx1h* mutant hearts (*Figure 10C*).

## Conclusion

Compromised $Ca^{2+}$ homeostasis and damaged cardiac muscle fibers are often observed in deteriorating diseased hearts, but a causative relationship between these outcomes has not previously been demonstrated. In this study, we used the zebrafish *ncx1h* mutant as an animal model to explore the molecular link between $Ca^{2+}$ signaling and myofibril integrity in the heart. We showed that NCX1 activity is dispensable for the initial assembly of sarcomeres, but the maintenance of myofibril structure in myocardial cells requires tightly controlled $Ca^{2+}$ homeostasis and MuRF1 expression.

Our molecular analyses using cultured cells and in vivo studies in zebrafish reveal a FoxO-MuRF1 signaling axis that is critically involved in the $Ca^{2+}$-dependent regulation of myofibril integrity in the heart. We propose that under normal physiological conditions where the cytosolic diastolic $Ca^{2+}$

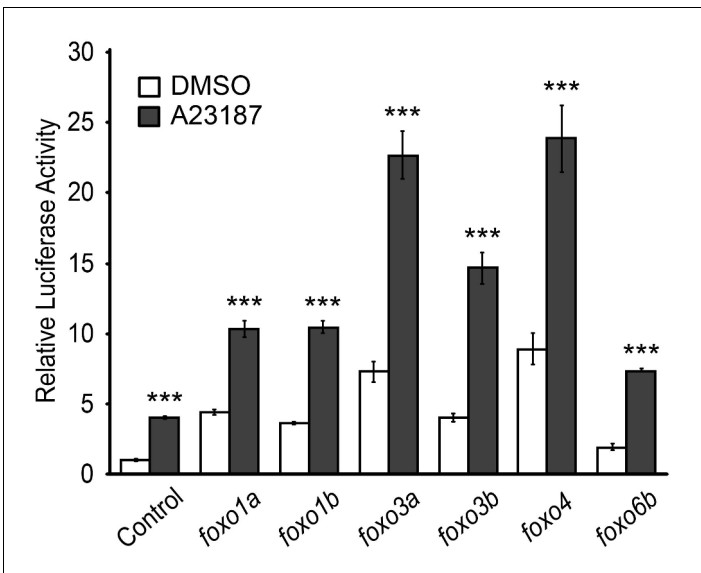

**Figure 9.** FoxO induces MuRF1 expression. Luciferase activity of the MuRF1a (−638) reporter cotransfected with different *foxo* genes. Values on the y-axis are expressed relative to the luciferase activity of DMSO treated cells. ***p<0.001.

DOI: https://doi.org/10.7554/eLife.27955.013

level is low, FoxO is sequestered in the cytoplasm and MuRF1 expression is maintained at a basal level to support the normal turnover of sarcomeric proteins. Under pathological conditions, when diastolic $Ca^{2+}$ is elevated, the activation of Cn dephosphorylates FoxO and allows its nuclear translocation, leading to upregulation of MuRF1 and the degradation of myofibrils (*Figure 10D*). Interfering with the Cn-FoxO-MuRF1-proteosome pathway by pharmacological or genetic means can protect the sarcomeric integrity of cardiomyocytes suffering from $Ca^{2+}$ dysregulation, indicating that the FoxO-MuRF1 signaling axis is a central regulator of the $Ca^{2+}$-dependent growth and degradation of striated muscles. The activity of the Cn-FoxO-MuRF1 signaling pathway identified in this study is consistent with the roles of the FoxO-MuRF1 pathway in hypertrophy and atrophy responses in skeletal muscles (*Sacheck et al., 2004*; *Stitt et al., 2004*; *Waddell et al., 2008*) and suggests that FoxO-MuRF1 signaling is critical to the maintenance of tissue homeostasis and the response of myocytes to pathological stimuli. Furthermore, cardiac-specific overexpression of MuRF1 results in phenotypes resembling those observed in cardiomyopathy, including the breakdown of sarcomeres and a dilated heart with reduced heart rate and decreased contractility, raising the possibility that misregulation of MuRF1 contributes to the pathological progression of cardiovascular diseases. Interestingly, cardiac patients carrying specific *murf1* gene variants have a poor prognosis (*Chen et al., 2012*; *Su et al., 2014*), suggesting that MuRF1 has a conserved role in the regulation of cardiac structure and function from lower vertebrates to humans and raising an intriguing possibility that the Cn-FoxO-MuRF1-proteosome pathway may be an attractive point of therapeutic intervention for cardiomyopathies. The complete loss of *ncx1h* activity in *tremblor* mutant cardiomyocytes eliminates $Ca^{2+}$ cycling and causes embryonic lethality, an extreme condition that is more severe than what is observed in patients with chronic heart disease (*Ebert et al., 2005*; *Langenbacher et al., 2005*). Future studies using clinically relevant mammalian models will further illuminate the therapeutic potential of targeting the Cn-FoxO-MuRF1-proteosome pathway in the context of cardiovascular disease.

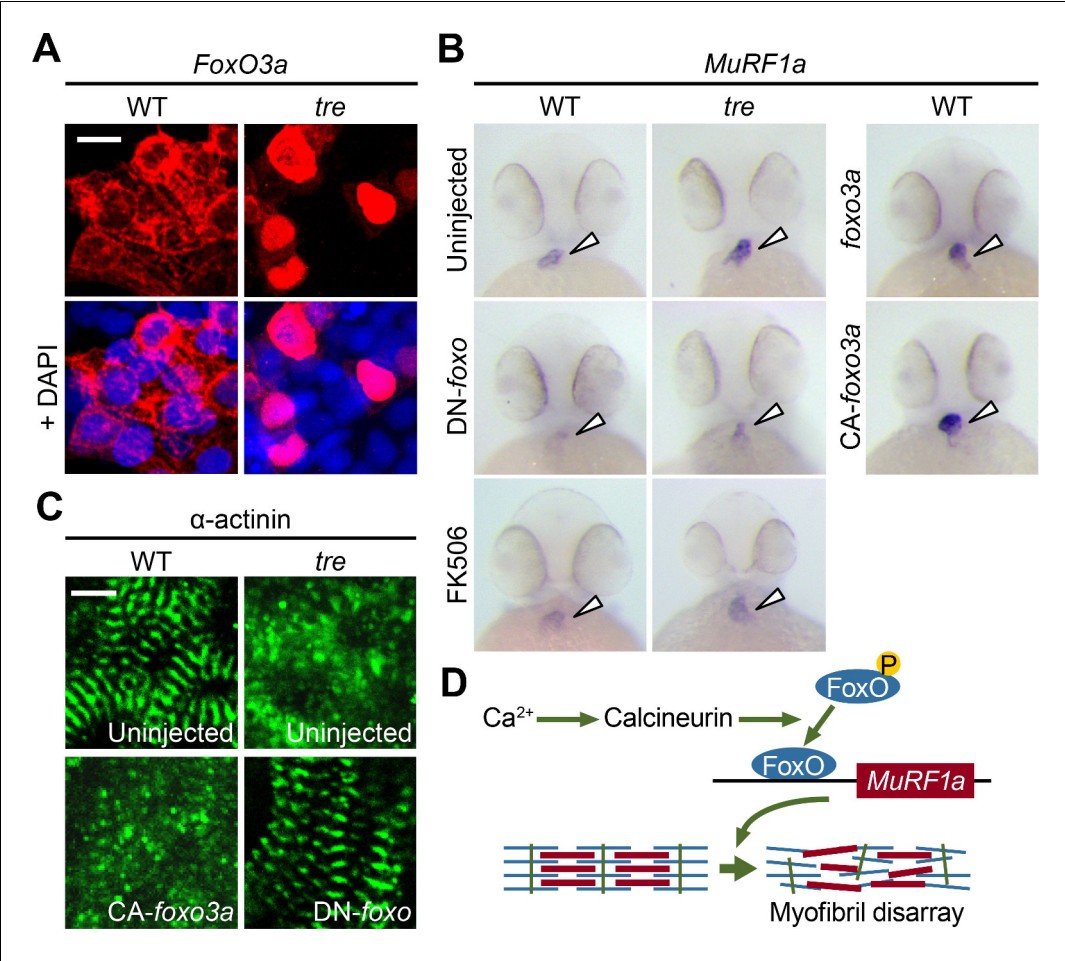

**Figure 10.** FoxO3a regulates MuRF1 expression in the heart. (**A**) FoxO3a is predominantly localized in the cytoplasm of 2-day-old wild type cardiomyocytes, but is concentrated in the nuclei of *tre* cardiomyocytes. FoxO3a is pseudo colored in red and nuclei are labeled by DAPI in blue. Scale bar: 10 µm. (**B**) In situ hybridization showing stronger MuRF1 signals in *tre* mutant and CA-foxo3a injected two dpf hearts compared to wild type siblings. *Murf1a* expression in *tre* hearts is suppressed by DN-foxo overexpression or FK506 treatment. (**C**) Immunostaining of α-actinin in three dpf hearts. Intact sarcomeres were detected in control (uninjected) and DN-foxo3a injected *tre* hearts whereas disassembled sarcomeres were observed in *tre* and CA-foxo3a injected hearts. Scale bar: 5 µm. (**D**) Model for Ca²⁺ overload-induced myofibril disarray. Calcineurin dephosphorylates FoxO leading to FoxO nuclear translocation, MuRF1 expression and sarcomere disassembly.
DOI: https://doi.org/10.7554/eLife.27955.014

## Materials and methods

### Zebrafish husbandry, chemical treatment and morpholino knockdown

Zebrafish *tremblor* $^{tc318}$ heterozygotes were bred in the *Tg(myl7:EGFP)* background and raised as previously described (*Westerfield, 2000*). Embryos were raised at 28.5°C and staged as previously described (*Kimmel et al., 1995*). For Cn or proteasome inhibition, embryos were treated with 10 µM FK506 (Sigma-Aldrich, St. Louis, MO) or 50 µM MG132 (Sigma-Aldrich) at 24 hpf. The morpholino-modified antisense oligonucleotides targeting the translation initiation sites of *murf1a* and *murf1b* (*Table 1*, Gene Tools) were microinjected at the 1- to 2 cell stage (8 ng each). This study was performed in strict accordance with the recommendations in the Guide for the Care and Use of Laboratory Animals of the National Institutes of Health. All of the animals were handled according to approved institutional animal care and use committee (IACUC) protocols of the University of

**Table 1.** Primers and morpholinos used in this manuscript

| Experiment | Target gene | Sequence |
|---|---|---|
| Quantitative RT-PCR | murf1a | F: 5'- GGAAGAAAACTGCCAGGCACAG −3'<br>R: 5'- CTGGGTGATCTGCTCCAGAAGATG −3' |
| | murf1b | F: 5'- CAGGACAATGCTCAACGTGCC −3'<br>R: 5'- CTTGCTCTTTGCCAATACGCTCTAAGAG −3' |
| Molecular cloning* | murf1a | F: 5'- CTGA<u>GGTACC</u>AAGCAGTGAAGGTTA −3'<br>R: 5'- GCTA<u>GGTACC</u>AGTCTCTCATTGCTT −3' |
| | murf1b | F: 5'- CTAT<u>GGATCC</u>CTGCAGGGAATCATTTAC −3'<br>R: 5'- GTTA<u>CTCGAG</u>CATTTGTCAATGACCTTG −3' |
| | foxo1a | F: 5'- GTCT<u>GAATTC</u>CAGTATTGCTGGTACCATG −3'<br>R: 5'- CATT<u>GCTAGC</u>ACTACCCAGACACCCAG −3' |
| | foxo1b | F: 5'- GTAT<u>GGATCC</u>TTGGTGATGGCAGAACC −3'<br>R: 5'- GTAT<u>CTCGAG</u>CAGCAGATGACATGTCTATC −3' |
| | foxo3a | F: 5'- GTAT<u>GGATCC</u>GGAGTCGAGGAAATATGG −3'<br>R: 5'- GTAT<u>CTCGAG</u>CAGTTGCTTTACAGTGGAC −3' |
| | foxo3b | F: 5'- GTAT<u>GGATCC</u>CGACCAAGACAGTAAAGAG −3'<br>R: 5'- GTAT<u>CTCGAG</u>CTGAGCAATTCCCATCAG −3' |
| | foxo4 | F: 5'- GTCT<u>GAATTC</u>CATCGCACAATGGAGG −3'<br>R: 5'- CATT<u>GCTAGC</u>CAACAGTGGAGTTAGCT −3' |
| | foxo6a | F: 5'- ATGA<u>GGATCC</u>AACTCCATTAGACACAACC −3'<br>R: 5'- GCTA<u>GAATTC</u>GTGTGATTGTTGAGGTCC −3' |
| | foxo6b | F: 5'- ATGA<u>GGATCC</u>CGGTTTCTTAAGCACAGAAG- 3'<br>R: 5'- ATGA<u>GAATTC</u>GACATTTATCCAGGCACC- 3' |
| | MuRF1a(−6906)<br>MuRF1a(−639)<br>MuRF1a(−582)<br>MuRF1a(−460)<br>MuRF1a(−38)<br>Common Reverse primer for MuRF1 | F: 5'- GTTA<u>GCTAGC</u>CGACTTACTCACTCC −3'<br>F: 5'- GTACTTGGA<u>GCGGCCGC</u>AATAA −3'<br>F: 5'- GTCA<u>GCTAGC</u>CCAACCCAGACAATATATTACT −3'<br>F: 5'- GTCG<u>GCTAGC</u>GGGAAATAATAATATTGTGATTG −3'<br>F: 5'- GACT<u>GCTAGC</u>CGGCTGGTATATAAGAC −3'<br>R: 5'- GAAT<u>CTCGAG</u>TGCTGAGGTAGAGTC −3' |
| Morpholino | Control<br>murf1a<br>murf1b | 5'-CCTCTTACCTCAGTTACAATTTATA-3'<br>5'-TTTGACCCGTTTGGATGTCCATTGC-3<br>5'-AAGAGGCAGTTCGCTGAATGTCCAT-3' |

*Restriction enzyme sites are underlined.

DOI: https://doi.org/10.7554/eLife.27955.015

California, Los Angeles. The protocol was approved by the Chancellor's Animal Research Committee of the University of California, Los Angeles.

## Zebrafish transgenesis

Transgenic constructs, *myl7:MuRF1a-IRES-EGFP* and *myl7:FLAG-foxo3a-IRES-EGFP*, were generated using the Tol2kit (*Kwan et al., 2007*). Wild type embryos were injected at the 1 cell stage with 10–20 pg of the transgene plasmid and 20 pg of mRNA encoding Tol2 transposase. Embryos with cardiac-specific EGFP expression were raised as founders.

## Microarray and quantitative PCR

Wild type and *tre* mutant hearts were isolated at 48 hpf as previously described (*Geoffrey Burns and MacRae, 2006*). Total RNA was purified using the RNeasy micro kit (Qiagen, Valencia, CA). Microarray hybridization was performed in triplicate using the Affymetrix Zebrafish GeneChip containing 15,617 genes. Data were analyzed using scripts written in the statistical programming language R (*R Development Core Team, 2014*). Differentially expressed genes were identified using linear models and multiple testing correction implemented in the Limma package (*Smyth, 2004*). The relative expression levels of *murf1a* and *murf1b* in the wild type and *tre* hearts were determined by quantitative PCR using the LightCycler 480 System (Roche Applied Science). GAPDH served as the internal control for normalization. Primer sequences used in this study are listed in *Table 1*.

### In vivo GFP reporter assay

An approximately 7.0 kb genomic fragment upstream of the zebrafish *murf1a* gene (ranging from −6906 to +80 bp) was amplified from genomic DNA (*Table 1*). A deletion series of *MuRF1a*-EGFP construct was generated using the ERASE-A-BASE system (Promega, Madison, WI). For transient expression analysis, each deletion construct was digested with NheI and SalI to release the *MuRF1a*-EGFP reporter and microinjected into 1 cell stage embryos. A minimum of 20 EGFP-positive embryos of each group were examined at 1 and 2 days post fertilization using a Zeiss SV-11 epifluorescence microscope.

### Whole-mount in situ hybridization, immunostaining and Western analysis

Whole mount in situ hybridization, immunostaining and Western analysis were performed as previously described (*Langenbacher et al., 2011*; *Cavanaugh et al., 2015*). The antisense RNA probes were synthesized from pCS2 +expression constructs containing a partial genomic fragment (*foxo5a*) or full-length cDNA fragments (*murf1a, murf1b, foxo1a, foxo1b, foxo3a, foxo3b, foxo4,* and *foxo5b*) (*Table 1*). Goat anti-MuRF1 (1:500 dilution, R and D Systems, AF5366) and rabbit anti-goat-HRP (1:15,000 dilution, Thermo Fisher Scientific, 81–1620) were used for Western analysis. Phalloidin (1:50, Sigma-Aldrich), anti-sarcomeric α-actinin (1:1000, clone EA53, Sigma-Aldrich), α-FLAG (1:100, clone M2, Sigma-Aldrich) and Zn8 (1:100, Developmental Studies Hybridoma Bank, Iowa City, IA) were used for immunostaining. Fluorescence images were acquired using an LSM 510 confocal microscope (Zeiss, Germany) with a 40x water objective. Embryos were classified as having intact sarcomeres if they exhibited at least five adjacent, clearly defined Z-lines marked by α-actinin in any area of the ventricle.

### Cardiac imaging and analysis

Videos of *Tg(myl7:MuRF1-IRES-EGFP)* and *Tg(myl7:EGFP)* hearts were taken at 30 frames per second. Cardiac parameters were assessed by line-scan analysis as previously described (*Shimizu et al., 2015*).

### Cell-based luciferase assay

HEK293T cells (ATCC, Manassas, VA) were plated into 96-well plates at a density of 32000 cells per well and transfected with 200 ng of the *MuRF1a* (−6906)- or *MuRF1a* (−638)-luciferase reporter construct, 50 ng of the SV40-*Renilla* luciferase reporter construct and expression vectors (Cn, DN-Cn, foxo3a, CA-foxo3a or DN-foxo3a). Cells were treated with 5 μM A23187 (Sigma-Aldrich), 0.5 μM FK506 (Sigma-Aldrich) or 0.5 μM KN62 (Sigma-Aldrich). Luciferase activities were determined with the Dual-Glo Luciferase Assay System (Promega) in triplicate at least three times, and the activity of firefly luciferase was normalized to that of *Renilla* luciferase for transfection efficiency and cell viability. The identity of the HEK293T cell line has been authenticated by STR profiling using the Promega PowerPlexX16 System recommended by the American Type Culture Collection (Laragen, Inc., Culver City, CA). No mycoplasma contamination was detected (Laragen, Inc.).

### Statistics

Sample sizes with adequate statistical power were empirically determined based on previous experiments. Samples were randomly allocated into control and experimental groups. Experiments of each condition were performed at least three times on independent biological replicates. Results are presented as the mean ±S.E. *p*-values associated with all comparisons are based on unpaired two-sided Student's *t*-tests (n ≥ 3) unless otherwise stated. All data values were included in the analysis.

## Acknowledgements

The authors thank Dr. Chen Gao and members of the Chen Lab for stimulating discussions. This work was supported by grants from the Nakajima Foundation (to HS), the National Institute of Health (HL096980 and HL126051 to JNC and HL108186 to YW), European Commission's Sixth Framework Programme (ZF-MODELS project to RG) and Seventh Framework Programme (ZF-HEALTH project to RG).

## Additional information

### Funding

| Funder | Grant reference number | Author |
|---|---|---|
| National Institutes of Health | HL096980 | Jau-Nian Chen |
| European Commission | ZF-MODELS | Robert Geisler |
| European Commission | ZF-HEALTH | Robert Geisler |
| National Institutes of Health | HL126051 | Jau-Nian Chen |
| National Institutes of Health | HL108186 | Yibin Wang |
| Nakajima Foundation | | Hirohito Shimizu |

The funders had no role in study design, data collection and interpretation, or the decision to submit the work for publication.

### Author contributions

Hirohito Shimizu, Conceptualization, Data curation, Formal analysis, Validation, Investigation, Methodology, Writing—original draft, Writing—review and editing; Adam D Langenbacher, Data curation, Formal analysis, Investigation, Methodology, Writing—review and editing; Jie Huang, Data curation, Formal analysis, Methodology; Kevin Wang, Formal analysis, Methodology; Georg Otto, Data curation, Formal analysis, Validation; Robert Geisler, Data curation, Formal analysis, Supervision, Writing—review and editing; Yibin Wang, Supervision, Funding acquisition, Writing—review and editing; Jau-Nian Chen, Conceptualization, Data curation, Formal analysis, Supervision, Funding acquisition, Investigation, Methodology, Writing—original draft, Project administration, Writing—review and editing

### Author ORCIDs

Hirohito Shimizu http://orcid.org/0000-0002-9889-3892
Adam D Langenbacher http://orcid.org/0000-0002-3752-0208
Georg Otto http://orcid.org/0000-0002-3929-948X
Jau-Nian Chen http://orcid.org/0000-0001-8807-3607

### Ethics

Animal experimentation: This study was performed in strict accordance with the recommendations in the Guide for the Care and Use of Laboratory Animals of the National Institutes of Health. All of the animals were handled according to approved institutional animal care and use committee (IACUC) protocols of the University of California, Los Angeles. The protocol was approved by the Chancellor's Animal Research Committee of the University of California, Los Angeles (ARC#2000-051-51A).

### Decision letter and Author response

Decision letter https://doi.org/10.7554/eLife.27955.017
Author response https://doi.org/10.7554/eLife.27955.018

## Additional files

### Supplementary files

• Transparent reporting form
DOI: https://doi.org/10.7554/eLife.27955.016

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
