## [Decision Letter]

Thank you for submitting your article "The Calcineurin-FoxO-*Murf1* Signaling Pathway Regulates Myofibril Integrity in Cardiomyocytes" for consideration by *eLife*. Your article has been favorably evaluated by Didier Stainier (Senior Editor) and three reviewers, one of whom is a member of our Board of Reviewing Editors. The reviewers have opted to remain anonymous.

The reviewers have discussed the reviews with one another and the Reviewing Editor has drafted this decision to help you prepare a revised submission.

Summary:

In this manuscript, Shimizu et al. provide valuable insight into a potential mechanism connecting calcium overload and myofibril integrity. Prior studies in both mouse and zebrafish have shown that loss of function of the sodium-calcium exchanger Ncx1 results in myofibril disarray, but the link between this cause and effect has been unclear. Through an appealing combination of approaches both in zebrafish and in cell culture, the authors provide evidence for a model in which abnormal calcium handling in cardiomyocytes leads to calcineurin-mediated translocation of FoxO to the nucleus, upregulating *Murf1*, which then degrades myofibrils. The authors' experiments are cleverly designed with clear logic, and their results are nicely presented. Their conclusions advance our understanding of the mechanisms responsible for normal myofibril homeostasis and for responses to aberrant stimuli. Moreover, these findings inspire ideas for potential therapeutic approaches for cardiomyopathies.

While most results are reported clearly and explained thoroughly, a few elements are left unexplained are could be more strongly supported. Thus, several issues should be addressed in a revised manuscript, as detailed below.

Essential revisions:

1) Prior studies in zebrafish (from the Xu and Yelon labs) have implicated cardiac contractility in the regulation of sarcomere integrity. While it is clear that calcium is overloaded in ncx1 mutants, cardiac contractility is also compromised. Is it possible that the latter change could contribute to the myofibril integrity defects? How might it be possible to distinguish the relative contributions of reduced contractility and calcium overload? Is there any scenario that would allow analysis of the calcineurin-FoxO-*Murf1* pathway in the context of calcium overload with normal contractility (or is this impossible to achieve)? At a minimum, is it possible to demonstrate that the calcineurin-FoxO-*Murf1* pathway is not triggered by reduction of contractility (perhaps in silent heart mutants, or in blebbistatin treated embryos)? Or, since *murf1* morpholinos and MG132 can restore myofibril integrity in ncx1-deficient hearts, is it informative to examine cardiac rhythm in these rescued hearts?

2) Another important issue relates to Ca^2+^ compartmentalization. Distinct patterns of Ca^2+^ signals are generated in space and time to stimulate specific cellular responses. Is the Ca^2+^ that controls muscle contraction also involved in activating calcineurin?

3) Figure 4: Comparing the ventricular morphology in Figure 1/3A vs. Figure 4, the ventricle in Tg:*murf1* embryos was relatively normal in comparison to what is seen in a ncx1 mutant. Does Tg:*murf1* fully recapitulate the phenotype of the ncx1 mutant? Quantification of the level of overexpression of the transgene is likely to help with the interpretation of these data. Along the same lines, the authors state that *murf1* overexpression led to impaired cardiac function. However, Figure 4 only shows a reduced atrial fraction shortening, instead of ventricular fraction shortening. Can the authors provide an assessment of ventricular performance in Tg:*murf1* embryos?

4) Figure 10: Figure 10 demonstrates that FoxO translocates into the nuclei of some ncx1 mutant cardiomyocytes, yet it seems that many of the ncx1 mutant cardiomyocytes exhibit a decreased level of FoxO or no FoxO. Please clarify.

5) Validation of reagents: A pair of morpholinos are employed in Figure 5 to knockdown murf1a and murf1b function. The sequences of the morpholinos are provided in Table 1, but no validation of these reagents is described. Considering that the morpholinos improve sarcomere integrity in the ncx1 mutants, there is no major concern that they are causing general toxicity, but it would be helpful to describe whatever controls the authors performed for these tools.

6) Validation of reagents: Figure 10 employs a FoxO antibody, but there is no description of this reagent in the Materials and methods. Where does this antibody come from, and how has it been validated?

---

## [Author Response]

*Essential revisions:*

*1) Prior studies in zebrafish (from the Xu and Yelon labs) have implicated cardiac contractility in the regulation of sarcomere integrity. While it is clear that calcium is overloaded in ncx1 mutants, cardiac contractility is also compromised. Is it possible that the latter change could contribute to the myofibril integrity defects? How might it be possible to distinguish the relative contributions of reduced contractility and calcium overload? Is there any scenario that would allow analysis of the calcineurin-FoxO-Murf1 pathway in the context of calcium overload with normal contractility (or is this impossible to achieve)? At a minimum, is it possible to demonstrate that the calcineurin-FoxO-Murf1 pathway is not triggered by reduction of contractility (perhaps in silent heart mutants, or in blebbistatin treated embryos)? Or, since murf1 morpholinos and MG132 can restore myofibril integrity in ncx1-deficient hearts, is it informative to examine cardiac rhythm in these rescued hearts?*

To address the reviewer’s question, we provide a new video showing that *ncx1h/murf1a/murf1b* triple deficient hearts exhibit a chaotic fibrillation-like movement similar to *ncx1h* hearts (Video 3). This study demonstrates that knockdown of *Murf1* preserves sarcomere integrity without restoring cardiac contraction in *ncx1h* deficient embryos, indicating that the regulation of contractility and sarcomere integrity can be uncoupled (subsection “Blocking *MuRF1*-induced protein degradation preserves myofibril integrity in *ncx1h* mutant hearts”).

*2) Another important issue relates to Ca^2+^ compartmentalization. Distinct patterns of Ca^2+^ signals are generated in space and time to stimulate specific cellular responses. Is the Ca^2+^ that controls muscle contraction also involved in activating calcineurin?*

We apologize for the confusion. In this revised manuscript, we have stated that both *ncx1h* deficiency (Introduction) and A23187 treatment (subsection “Ca^2+^ induces *MuRF1a* expression”) result in a global Ca^2+^ homeostasis defect in affected cells.

*3) Figure 4: Comparing the ventricular morphology in Figure 1/3A vs. Figure 4, the ventricle in Tg:murf1 embryos was relatively normal in comparison to what is seen in a ncx1 mutant. Does Tg:murf1 fully recapitulate the phenotype of the ncx1 mutant? Quantification of the level of overexpression of the transgene is likely to help with the interpretation of these data. Along the same lines, the authors state that murf1 overexpression led to impaired cardiac function. However, Figure 4 only shows a reduced atrial fraction shortening, instead of ventricular fraction shortening. Can the authors provide an assessment of ventricular performance in Tg:murf1 embryos?*

*Murf1* overexpression recapitulates the sarcomeric disassembly phenotype present in *tre/ncx1h* hearts, but does not induce the severe morphological defects that are likely attributable to the complex cellular responses induced by aberrant Ca^2+^ handling and the failure of *tre/ncx1h hearts* to initiate a heartbeat (Langenbacher et al., 2005, Shimizu et al., 2015).

Overexpression of *Murf1* affects both atrial and ventricular performance (new Video 1 and Video 2). The quantitative data on ventricular fractional shortening (VFS) is now presented in the text (subsection “*MuRF1* regulates myofibril integrity in cardiomyocytes”).

*4) Figure 10: Figure 10 demonstrates that FoxO translocates into the nuclei of some ncx1 mutant cardiomyocytes, yet it seems that many of the ncx1 mutant cardiomyocytes exhibit a decreased level of FoxO or no FoxO. Please clarify.*

FoxO is regulated by its subcellular localization. As current FoxO antibodies are not suitable for whole mount immunostaining in zebrafish, we injected a small amount of DNA encoding FLAG-tagged FoxO3a into zebrafish embryos and used the FLAG tag as a proxy to assess the subcellular localization of FoxO. This procedure creates a mosaic heart with a small number of cardiomyocytes expressing FLAG-tagged FoxO. We have revised the text to clarify the experimental design (subsection “Cn and FoxO regulate *MuRF1* expression in the heart”).

*5) Validation of reagents: A pair of morpholinos are employed in Figure 5 to knockdown murf1a and murf1b function. The sequences of the morpholinos are provided in Table 1, but no validation of these reagents is described. Considering that the morpholinos improve sarcomere integrity in the ncx1 mutants, there is no major concern that they are causing general toxicity, but it would be helpful to describe whatever controls the authors performed for these tools.*

In addition to examining sarcomeric integrity in *tre/ncx1h* mutant hearts that received control morpholino, we assessed the amount of *Murf1* protein in control embryos and morphants by Western blotting. We found that control MO cannot preserve the sarcomeric structure in *ncx1h* deficient hearts (New Figure 5) and that the protein level of *Murf1* is reduced by 27% in *Murf1* morphants (New Figure 5, and subsection “Blocking MuRF1-induced protein degradation preserves myofibril integrity in *ncx1h* mutant hearts”).

*6) Validation of reagents: Figure 10 employs a FoxO antibody, but there is no description of this reagent in the Materials and methods. Where does this antibody come from, and how has it been validated?*

As explained above, we used the *myl7:FLAG-foxo3a-IRES-EGFP* transgene as a proxy to assess the subcellular localization of FoxO. For this purpose, we used an anti-FLAG antibody for immunostaining (Sigma-Aldrich). This information is presented in the “Materials and methods” section.